# Genome-Wide Identification and Characterization of *MYB* Transcription Factors in Sudan Grass under Drought Stress

**DOI:** 10.3390/plants13182645

**Published:** 2024-09-21

**Authors:** Qiuxu Liu, Yalin Xu, Xiangyan Li, Tiangang Qi, Bo Li, Hong Wang, Yongqun Zhu

**Affiliations:** Institute of Agricultural Resources and Environment, Sichuan Academy of Agricultural Sciences, Chengdu 610066, China; sicauliuqiuxu@163.com (Q.L.); xylxuyaling@163.com (Y.X.); lxy18384259655@163.com (X.L.); 17863660603@163.com (T.Q.); libo981811@163.com (B.L.); wang.hongde163@163.com (H.W.)

**Keywords:** Sudan grass, drought stress, genome-wide analysis, *MYB* transcription factors

## Abstract

Sudan grass (*Sorghum sudanense* S.) is a warm-season annual grass with high yield, rich nutritional value, good regeneration, and tolerance to biotic and abiotic stresses. However, prolonged drought affects the yield and quality of Sudan grass. As one of the largest families of multifunctional transcription factors in plants, MYB is widely involved in regulating plant growth and development, hormonal signaling, and stress responses at the gene transcription level. However, the regulatory role of *MYB* genes has not been well characterized in Sudan grass under abiotic stress. In this study, 113 *MYB* genes were identified in the Sudan grass genome and categorized into three groups by phylogenetic analysis. The promoter regions of *SsMYB* genes contain different cis-regulatory elements, which are involved in developmental, hormonal, and stress responses, and may be closely related to their diverse regulatory functions. In addition, collinearity analysis showed that the expansion of the *SsMYB* gene family occurred mainly through segmental duplications. Under drought conditions, *SsMYB* genes showed diverse expression patterns, which varied at different time points. Interaction networks of 74 *SsMYB* genes were predicted based on motif binding sites, expression correlations, and protein interactions. Heterologous expression showed that *SsMYB8*, *SsMYB15*, and *SsMYB64* all significantly enhanced the drought tolerance of yeast cells. Meanwhile, the subcellular localization of all three genes is in the nucleus. Overall, this study provides new insights into the evolution and function of *MYB* genes and provides valuable candidate genes for breeding efforts in Sudan grass.

## 1. Introduction

Sudan grass (*Sorghum sudanense* (Piper) Stapf.), belonging to the genus Sorghum of the family Gramineae, is a warm-season annual herbaceous and C4 plant with diploid chromosomes (2n = 2x = 20). It originated in the Sudan in Africa and is known for its broad adaptability, strong tillering ability, high yield, rich nutritional value, low cyanide content, and good taste [1,2,3]. The rapid regeneration and strong resilience make Sudan grass ideal for multiple green fodder, silage, hay preparation, or grazing [2,4]. Compared to other crops, pasture grasses are usually grown under more severe climatic conditions, significantly impacting their growth and development [5]. Drought stress not only leads to lower forage production but is also a major cause of economic losses in grassland ecology and livestock husbandry [6]. Therefore, the breeding of Sudan grass for drought tolerance is of significance for its production.

In response to harsh environmental conditions, higher plants have evolved a complex and fine-grained signaling network comprising several elements, including transcription factors (TFs). As the core elements of gene expression regulation, TFs play a crucial role in plant growth and development as well as in response to various abiotic stresses [7,8,9]. MYB, one of the largest families of TFs in plants, contains a highly conserved DNA binding domain (DBD) [10], which enables their specific binding to the DNA sequences of target downstream genes [11,12]. Therefore, MYB TFs can precisely regulate the expression of target genes and thus play a key role in plant growth, development, and response to environmental adversity. In 1987, the first plant *MYB* gene, *COLORED1*, encoding the protein ZmMYBC1, was cloned in maize (*Zea mays*) [13]. Since then, members of the MYB family have been identified in various plants, including *Arabidopsis* (*Arabidopsis thaliana*) [14], rice (*Oryza sativa*) [15], soybean (*Glycine max*) [16], and sugar beet (*Beta vulgaris*) [17]. Based on its similarity to the three repeat sequences R1, R2, and R3 in animal C-MYB and the number of R repeat sequences, the plant *MYB* family was classified into four categories, including 1R-MYB, R2R3-MYB, 3R-MYB, and 4R-MYB, of which R2R3-MYB is the most abundant subclass in the plant MYB family [18].

Plants inevitably encounter various adversities during development and have evolved a series of effective defense mechanisms to adapt to these adversities. Among these mechanisms, MYB TFs, which are associated with plant tolerance to abiotic adversity, have been widely reported. For example, *IbMYB48* overexpression significantly increased soluble sugar and proline contents and maintained the osmotic potential balance and membrane integrity inside and outside plant cells, therefore enhancing drought tolerance in sweet potato (*Ipomoea batatas*) [19,20]. *MdMYB46* positively regulates lignin accumulation and thus improves drought tolerance in apple (*Malus domestica*) [21]. Furthermore, the expression of *PtoMYB142* was induced by drought stress, enhancing the transcript abundance of the epidermal wax synthesis genes CER4 and KCS, leading to a significant reduction in water loss, and consequently increasing the drought tolerance of hairy poplar (*Populus tomentosa*) [22]. In recent years, many studies have shown that *MYB* genes are also widely involved in plant response to other abiotic stresses. For example, *MdMYB44-like*, which is highly expressed in the epidermal guard cells of apple leaves, can directly bind the MBS element in the promoter region of *MdPP2CA* and repress its transcriptional process The interaction of *MdMYB44-like* with the ABA receptor protein MdPYL8 dramatically enhances this transcriptional repression, and the plant rapidly accumulates ABA in response to salt stress [23]. Overexpression of *AtMYB5* conferred tolerance to high-temperature stress in transgenic *Arabidopsis* [24]. *MdMYB308L* could interact with *MdbHLH33* to enhance cold tolerance in apples [25]. In addition, as a positive regulator of aluminum tolerance, *AtMYB103* activates the transcription of the gene responsible for cell wall xyloglucan-O-acetylation modification (trichome birefringencelike27, TBL27) and reduces cell wall binding to aluminum ions, thereby significantly enhancing aluminum tolerance in transgenic *Arabidopsis* [26]. These results suggest that MYB transmits signals through downstream regulation or upon interaction with upstream and downstream factors as well as integrating multiple signaling pathways in response to various abiotic stresses.

In this study, the Sudan grass *MYB* gene family was explored in depth through comprehensive genomic analysis. The study focused on understanding the phylogenetic relationships, chromosomal localization, collinearity, diversity of gene structures, characterization of conserved motifs, and cis-regulatory elements in the promoter regions of the members of the Sudan grass *MYB* gene family. The expression patterns of *SsMYBs* genes were explored using the available transcriptomic data. Additionally, the interaction networks of *SsMYB8*, *SsMYB15*, and *SsMYB64* were evaluated to determine their roles in response to drought stress. The findings not only provide new perspectives for understanding the functions of the *MYB* gene family in Sudan grass but also provide a valuable theoretical basis for future Sudan grass improvement.

## 2. Results

### 2.1. Identification and Physicochemical Properties of MYB Members in Sudan Grass

A total of 113 *MYB* genes were identified in the Sudan grass genome, all of which possess the MYB-DNA domain (PF00249). These genes were provisionally labeled *SsMYB1*-*SsMYB113*, reflecting their order in the genome. The *SsMYB* genes exhibited a wide range of amino acid lengths, from 161 to 1034 aa. Their protein isoelectric points (PI) also varied from 4.57 to 11.45, while their molecular weights spanned from 18.02 kDa to 114.36 kDa. Predictive analyses of subcellular localization revealed that 112 SsMYB proteins were localized in the nuclear, and 1 in the cytoplasm. This subcellular distribution suggests a widespread biological role of transcriptional regulation of the *MYB* gene family in Sudan grass [27]. Detailed information is cataloged in Appendix A.

### 2.2. Phylogenetic Analysis of SsMYB Genes in Sudan Grass

Phylogenetic analyses were performed to investigate the evolutionary relationships between Sudan grass and members of the *Arabidopsis* and sorghum MYB families. This analysis included 72 *AtMYBs* from *Arabidopsis* and 69 *SbMYBs* from sorghum, as illustrated in Figure 1. All MYBs were categorized into 1R-MYB, R2R3-MYB, and 3R-MYB. According to a previous report [18], R2R3-MYBs were classified into twenty-five subgroups (S1–S25). These subfamilies include some R2R3-MYB members that did not originally belong to the S25 subfamily, and most of the subfamilies include R2R3-MYB members from Sudan grass and *Arabidopsis*, suggesting that some of the R2R3-MYB homologous members may have been lost during a long evolutionary process [18,28]. The R2R3-*MYB* gene family in plants typically contains more than 100 members, indicating its functional diversity. In addition, no 4R-MYB family genes were found in Sudan grass.

### 2.3. Multiple Sequence Alignment of SsMYB Genes

The *MYB* gene family, characterized by the MYB domain that typically comprises 1–4 incomplete repeats, is one of the largest families of TFs, playing pivotal roles in various plant biological processes. In this study, we performed a comprehensive multisequence alignment of the *MYB* gene family in Sudan grass to explore their evolutionary trajectories and functional diversification. Based on the conserved amino acid residues of the incompletely repeated sequences, we divided three subdomains (I–III) (Appendix A). Almost all MYB proteins contained three conserved sub-structural domains at the N-terminus of amino acids, indicating that they were highly conserved during evolution.

### 2.4. Gene Structure, Motif, and Cis-Element Analysis of SsMYB Genes

To comprehensively investigate the structural configuration, motif composition, and cis-regulatory elements of the *SsMYB* genes and determine their functional diversity and regulatory complexity, we built a phylogenetic tree of the *SsMYBs* and identified 10 different conserved motifs using the MEME tool (Figure 2 and Appendix A). The conserved amino acid sequences for each motif are presented in Appendix A. The results show that motif 1, motif 2, motif 3, and motif 5 are widely distributed in most SsMYB proteins, whereas the other motifs (e.g., motif 6, motif 7, motif 8, motif 9, and motif 10) are present only in some members. In addition, 81 SsMYB proteins contain motif-4 at the N-terminus (Figure 2b). Meanwhile, gene structure analysis showed that the 113 *MYB* genes had 0–13 introns and 1–14 exons (Figure 2d). Genes in the same subfamily had similar exon length, intron position, and number, indicating a structural conservation among members of the *MYB* gene family. Intriguingly, the lengths of these introns and exons varied widely, suggesting potential functional diversity [29].

To investigate the potential regulatory mechanisms of *SsMYB* gene expression, we conducted an analysis of the 2000 bp upstream promoter sequences of all identified *SsMYB* genes using the PlantCARE database (Figure 2c). Our findings revealed the presence of cis-acting regulatory elements associated with light, hormonal, stress, and developmental responses in the promoters of 113 *SsMYB* genes (Appendix A). Notably, light-responsive elements (TCCC-motif) were the most prevalent. Within the hormonal category, elements responsive to abscisic acid (ABRE), methyl jasmonate (MeJA, TGACG-motif), auxin (AuxRR-core), salicylic acid (TCA), and gibberellin (GARE_motif) were prominent. Stress-related elements primarily included low-temperature response (LTR) and defense and stress (TC-rich repeats). Furthermore, certain *SsMYB* promoters contained wound-responsive elements (WUN-motif). These findings indicate that *SsMYBs* exhibit responsiveness to various hormones, growth, development, and stress.

### 2.5. Chromosomal Distribution and Collinearity Analysis of MYB Genes in Sudan Grass

The diversification of gene clusters is propelled by unique duplication events, which are the key catalysts in the evolutionary process of various species. *SsMYB* chromosome mapping was conducted using TBtools, and the results showed that the 113 *SsMYBs* were unevenly distributed on the 10 chromosomes of Sudan grass (Figure 3). Chromosome 2 had the highest number of *MYBs* (25), followed by chromosome 1 (16), and chromosomes 3 and 5 (13).

The MCScanX method was used to analyze gene duplication events in Sudan grass and 21 gene pairs (21 segmental duplication gene pairs and 0 tandem duplication gene pairs) with gene duplication events were identified (Figure 4 and Appendix A). These results suggested that segmental duplication was the main amplification mechanism of the *SsMYB* gene family [30]. Collinearity analysis can identify different species via evolutionary and kinship relationships [28]. To further explore the gene duplication of *SsMYBs* and infer their phylogenetic mechanisms, we selected sorghum for comparative analysis with Sudan grass. In Sudan grass, 106 members of the *SsMYB* gene family were homologous to the corresponding genes in sorghum (Figure 5). However, these homologous genes showed an uneven distribution pattern on their respective chromosomes, which may affect the functional expression and regulation of the genes.

### 2.6. Expression Patterns of SsMYB Genes under Drought Stress and Interaction Networks

The transcript levels of the 113 *SsMYBs* genes in Sudan grass leaves were analyzed in detail using previous RNA-seq data at different stages under drought stress [31]. After screening the genes with low expression levels, a total of 70 *SsMYB* genes were identified to have significant expression changes under drought stress (Figure 6). For example, the expression levels of *SsMYB1*, *SsMYB2*, *SsMYB54*, *SsMYB63*, *SsMYB97*, and *SsMYB109* generally showed a decreasing trend after 6 to 144 h of drought stress, whereas the expression levels of *SsMYB67*, *SsMYB100*, and *SsMYB101* showed an increasing trend. This finding showed that *SsMYB* family members may play different roles in plant adaptation to arid environments. Under drought stress conditions, *SsMYB9*, *SsMYB48*, *SsMYB58*, *SsMYB78*, and *SsMYB79* expression slightly increased, while *SsMYB95*, *SsMYB102*, and *SsMYB103* showed insignificant changes in expression. Overall, most of the *SsMYB* genes showed significant changes in their transcript levels under drought stress.

Furthermore, the STRING database analysis revealed that 34 of the 74 *SsMYB* genes were involved in interactions that form a complex network in response to drought stress (Figure 7 and Appendix A). In this network, specific genes such as *SsMYB15* and *SsMYB65* play key hub roles (Figure 7). The KEGG enrichment analysis was then performed based on the intersections obtained from expression data and motif binding site predictions. The results showed that *SsMYB* TFs were significantly enriched in metabolic pathways such as those involving the biosynthesis of amino acids, ubiquinone and other terpenoid-quinones, and butanoate metabolism (Appendix A). These findings provide valuable clues for a deeper understanding of the functions of *MYB* genes in plant response and adaptation to drought stress.

### 2.7. Overexpression of SsMYB8, SsMYB15, and SsMYB64 Improves Drought Tolerance in Yeast

Combining the expression levels of RNA-Seq data and protein interaction network analysis, we selected three genes, *SsMYB8, SsMYB15*, and *SsMYB64*, for preliminary functional validation. The coding sequences of these genes were cloned and expressed in *Saccharomyces cerevisiae* strains INVSC1. Positive INVSC1 yeast cells were cultured on Synthetic Galactose Minimal Medium without Uracil (SG-Ura) and supplemented with different concentrations of polyethylene glycol (PEG). Negative controls could grow in the absence of PEG (0 mM PEG 3350) and at lower concentrations of PEG (30, 60, and 90 mM PEG 3350), but not at higher concentrations of PEG (120 and 135 mM PEG 3350). In contrast, the experimental groups pYES2-NTB-*SsMYB8*, pYES2-NTB-*SsMYB15*, and pYES2-NTB-*SsMYB64* showed the ability to grow under all PEG concentration conditions tested (Figure 8). This result suggests that the *SsMYB8, SsMYB15*, and *SsMYB64* may play a positive role in plant response to drought stress.

### 2.8. Subcellular Localization of SsMYB8, SsMYB15, and SsMYB64

To investigate the subcellular localization of *SsMYB8*, *SsMYB15*, and *SsMYB64*, we fused their CDS sequences into CaMV35S::GFP vectors and transiently transformed the resulting recombinant vectors into tobacco leaf cells. The results showed that the GFP signals of the empty vector were present in the cytoplasm, cell membrane, and nucleus, and the GFP signals of *SsMYB8*, *SsMYB15*, and *SsMYB64* overlapped with the nuclear RFP signals, indicating that they were all present in the nucleus (Figure 9). This is consistent with the predicted results.

## 3. Discussion

MYB is one of the largest multifunctional TFs in plants. It is widely involved in regulating multiple processes at the transcription level, including plant growth and development, signal transduction of various phytohormones, and responses to abiotic and biotic stresses [8,10]. Sudan grass, known for its high protein content and palatability, is a valuable forage crop. However, its yield and quality are significantly compromised under prolonged drought conditions [4]. While *MYB* genes have been extensively studied in model plants such as *Arabidopsis* and rice, their characterization in Sudan grass remains incomplete. Therefore, studying *MYB* genes in Sudan grass will not only help to reveal their transcriptional regulation mechanism but also unravel their tolerance mechanisms under drought stress.

The present study identified 113 genes encoding MYB TFs in the Sudan grass genome. Phylogenetic analysis showed that these genes could be classified into three major groups (1R-MYB, R2R3-MYB, and 3R-MYB), and R2R3-MYB was divided into 25 subgroups. Compared with 198 *MYB* genes in *Arabidopsis* [32] and 210 *MYB* genes in sorghum [32], the number of *MYB* genes in Sudan grass was significantly lower and there were large differences in their open reading frame (ORF) lengths, molecular weights (MW), and isoelectric points (PIs) (Appendix A). These findings reflect the complexity and diversity of SsMYB proteins during evolution. Phylogenetic analyses also showed that most *SsMYBs* were closely related to *AtMYBs* and *SbMYBs*. For example, *Arabidopsis AtMYB73/77* negatively regulated hypocotyl elongation and lateral root development under UV light [33]. *AtMYB5* and *AtTT2* up-regulated the HSF2 expression level, enhanced the antioxidant enzyme activity, and improved *Arabidopsis* tolerance to high-temperature stress [24]. Sorghum *SbMYBHv33* negatively regulated the salt tolerance of sorghum and *Arabidopsis* by modulating the accumulation of reactive oxygen species and ions [34]. Additionally, *SbMYBAS1* overexpression significantly increased salt tolerance in *Arabidopsis* [32]. The role of these *SsMYB* genes in abiotic stresses may provide potential genetic resources for Sudan grass breeding.

The DBD structural domain of MYB is located at the N-terminus and consists of 1–4 incomplete MYB repeats [10,35]. Motifs 1, 2, 3, and 5 are highly conserved in almost all MYB proteins as components of the MYB conserved structural domains and are essential for the functional specificity of these TFs. Multiple genes (e.g., *SsMYB5* and *SsMYB16*) have different exon and intron structures despite having the same motif composition. Such differences may be caused by three main mechanisms: gain or loss of exons or introns, exonization or pseudo-exonization, and insertions or deletions [30]. In addition, based on the prediction of the cis-acting regulatory elements, motifs associated with light-responsive elements were the most abundant in *SsMYB* genes (Figure 2 and Appendix A). Light responsiveness has been shown to be associated with plant growth and development [36]. Hormone-responsive elements are closely related to stress responses, and abscisic acid plays a key role in plant responses to abiotic stresses such as drought and high salt [37]. MeJA is an important cellular hormone in various biological processes such as stress tolerance, leaf senescence, and seed germination [38]. Auxin is a multifunctional hormone that interacts with gibberellins to regulate embryogenesis, cell division, plant structure, and spatial localization in plants [39]. Some *SsMYB* gene promoters had cis-elements involved in abiotic stress response, such as LTR involved in low-temperature response [40] and TC-rich repeats involved in plant defense and stress response [41]. However, there were no cis-acting elements on the promoters of 42 *SsMYB* genes, probably due to deletions or mutations during the evolutionary process. Further functional studies of these promoter cis-elements may help to reveal the regulatory mechanism of *SsMYB* in the growth and development of Sudan grass under abiotic stress.

Gene duplication events drive the rapid expansion and evolution of species gene families through chromosome doubling. In Sudan grass, 21 segmental duplication gene pairs were identified, while no tandem duplication gene pairs were found, suggesting that the expansion of the *SsMYB* gene family might have occurred mainly through segmental duplication [30,42]. Analysis of interspecies covariance showed that 106 members of the *SsMYB* gene family sorghum were homologous to the corresponding genes in sorghum, suggesting that Sudan grass is closely related to sorghum.

Previous studies have shown that *MYB* genes are extensively involved in regulating stress responses at the transcriptional level. For example, the expression of *IbMYB48* increased under drought stress, and *IbMYB48* overexpression significantly enhanced drought tolerance in sweet potato (*Ipomoea batatas*) [19]. Buckwheat (*Fagopyrum tataricum*) *FtMYB10* self-expression was induced by ABA and drought stress, and *Arabidopsis* overexpressing *FtMYB10* showed reduced resistance to salt and drought. [43]. In this study, 43 *SsMYB* genes showed a decreasing trend under drought stress, while 27 *SsMYB* genes showed an increasing trend (Figure 6). Among them, the expression levels of *SsMYB8* and *SsMYB64* were significantly increased under drought stress at different time points (Figure 6). Subsequently, we performed STRING interaction network prediction and KEGG enrichment analysis through expression level analysis and motif binding site prediction. The results showed that *SsMYB15* and *SsMYB65* might play an important function as hub genes in the interworking network. In addition, abiotic stress can induce increases in various amino acids and ROS in plants [20], and KEGG analysis suggests that *SsMYB* may have potential applications in enhancing genetic improvement of drought tolerance in plants through ROS-scavenging pathways. In summary, we selected three genes, *SsMYB8*, *SsMYB15*, and *SsMYB64*, for preliminary functional validation of drought tolerance and successfully improved the drought tolerance of yeast (Figure 8). Although the current study revealed the response of *SsMYB8*, *SsMYB15*, and *SsMYB64* to drought stress, it is also important to evaluate the potential role of other *MYB* genes in mediating the plant stress response. Additionally, further studies are necessary to reveal the molecular mechanisms of drought tolerance in Sudan grass and to provide effective strategies for its future molecular breeding.

## 4. Materials and Methods

### 4.1. Identification and Bioinformatic Analysis of SsMYB Genes

For the identification and bioinformatic analysis of *SsMYB* genes, genomic resources and annotations for Sudan grass were obtained from NCBI (https://www.ncbi.nlm.nih.gov/bioproject/PRJNA830304/ (accessed on 17 January 2024)) [44]. The Hidden Markov Model (PF00249) profile of the MYB domain was retrieved from the Pfam Protein Family Database (http://Pfam.xfam.org/ (accessed on 17 January 2024)) and the local Sudan grass protein database was searched using HMMER (v3.3.2) (http://hmmer.janelia.org/ (accessed on 17 January 2024)) to characterize the *MYB* gene family [45]. Finally, potential SsMYB proteins were identified based on their conserved structural domains using the NCBI database (https://www.ncbi.nlm.nih.gov/Structure/cdd/wrpsb.cgi (accessed on 17 January 2024)) [46]. The physicochemical properties of SsMYB proteins, including amino acid length, relative isoelectric point (PI), and molecular weight (MW), were calculated using online Prosite ExPASy (https://www.expasy.org/ (accessed on 20 January 2024)) and SMART (http://smart.embl.de/smart/batch.pl (accessed on 20 January 2024)) servers [47,48]. Additionally, the subcellular localization of SsMYB proteins was predicted using the WolfPSORT (https://wolfpsort.hgc.jp/ (accessed on 20 January 2024)) database [49].

### 4.2. Phylogenetic Analysis and Multiple Sequence Alignment of SsMYBs

The MYB protein sequences of *Arabidopsis* and sorghum were obtained from the Ensembl Plant database (http://plants.ensembl.org/index.html (accessed on 30 January 2024)) [50]. Comparative analysis of all MYB protein sequences was conducted using clustalW (https://www.ebi.ac.uk/Tools/msa/clustalo/ (accessed on 30 January 2024)), and the data were saved in Mega format. [51]. Subsequently, MEGA7 software was utilized to construct a phylogenetic tree using the neighbor-joining (NJ) method with 1000 bootstrap replicates. The visualization and optimization of the phylogenetic tree were conducted via Evolview v3 (http://evolgenius.info/#/ (accessed on 31 January 2024)) [52], and the protein sequence comparison and editing of MYB conserved structural domains were performed using Jalview software v2.11.3.2 http://www.jalview.org/ (accessed on 31 January 2024)) [53].

### 4.3. Gene Structure, Motif, and Cis-Element Analysis

The exon and intron details for each *SsMYB* gene were extracted from the genome annotation files of Sudan grass. Conserved protein motifs were identified using the MEME Suite web server (https://meme-suite.org/ (accessed on 4 February 2024)), with parameters set to a maximum of 10 motifs and a width range of 5 to 200 residues [54]. Visualization of the exons, introns, conserved motifs, and structural domains of the *SsMYB* genes was performed using TBtools software v2,118. The PlantCARE database (http://bioinformatics.psb.ugent.be/webtools/plantcare/html/ (accessed on 4 February 2024)) was employed to examine the cis-acting elements within the genomic DNA sequences situated 2000 bp upstream of the transcription start site for each *SsMYB* gene [55].

### 4.4. Chromosomal Locations, Gene Duplication, and Collinear Relationship

The chromosomal positions of each *SsMYB* gene were extracted from the GFF3 file of the Sudan grass genome and visualized using MapChart software v2.32 (https://academic.oup.com/jhered/article/93/1/77/2187477 (accessed on 4 February 2024)) [56]. Meanwhile, the gene duplication events in the *SsMYB* gene family were analyzed using the MCScanX toolkit [57]. Additionally, inter-species collinearity analysis was performed and plotted using MCscan (python version) (https://github.com/tanghaibao/jcvi/wiki/MCscan-(Python-version), (accessed on 4 February 2024)) [58].

### 4.5. SsMYBs Transcriptome Analysis Based on RNA-Seq Data

Sorghum sudanense cv. Chuansu No. 1 used in this study was planted in the Sichuan Chinese Academy of Sciences. The seeds of Sudan grass were sterilized with 75% ethanol for 30 s, then with 10% sodium hypochlorite solution for 5 min and finally rinsed thrice with sterile distilled water. The seeds were spread in plastic cups (radius 6 cm, height 14 cm) filled with quartz for germination, set in controlled growth chambers under a 12 h photoperiod, a day/night temperature cycle of 25/23 °C and 75% relative humidity. Drought stress was applied to 30-day-old seedlings. Aboveground plant parts were collected at different drought stages (0, 6, 12, 24, 48, 72, and 144 h), as described previously [59]. RNA extraction from the collected samples was performed using TRIzol reagent (Life Technologies, Carlsbad, CA, USA) and Illumina sequencing was carried out using 1 μg of the RNA from each sample [59]. The experiment was conducted in three independent biologicals. Subsequently, a cDNA library was prepared using the NEBNext Ultra™ RNA Library Preparation Kit (NEB Corporation, Beverly, MA, USA), following the standard protocol of the manufacturer. The quantification of transcript abundance for the *SsMYB* gene was conducted in TPM (transcripts per million), and heatmaps were plotted based on log-transformed TPM values (log2(TPM + 0.01)) using the pheatmap package in the R language. Two metrics were used to screen for differentially expressed genes (DEGs), multiplicity of difference |Fold Change| ≥ 2 and FDR < 0.05.

### 4.6. SsMYBs Protein Interaction and KEGG Enrichment Analysis

Interaction prediction was performed based on motif binding sites, expression correlation, and protein interaction prediction (STRING) (http://string-db.org (accessed on 5 February 2024)) [60]. Motifs of target TFs were extracted from the upstream 2000 sequences of all genes for binding prediction analysis (with a threshold of 1 × 10^−7^). Based on the gene expression data, the Pearson correlation coefficient method was used to filter the set of genes with expression correlation of |cor| < 0.8 and *p* < 0.05. Finally, the protein sequences of *SsMYB* family genes as well as the gene sets obtained by the above two methods were loaded on the STRING database to evaluate the interactions between genes and map the protein interaction network using cytoscape. KEGG enrichment analysis was performed based on the intersections obtained from expression data and motif binding site prediction.

### 4.7. Heterologous Expression of SsMYB8, SsMYB15, and SsMYB64 in Yeast

The coding sequences (CDS) of *SsMYB8*, *SsMYB15*, and *SsMYB64* genes were PCR-amplified using homology arm primers containing *NdeI* and *BamHI* cleavage sites. These genes were subsequently cloned into the pYES2-NTB vector using the MonClone^TM^ Single Assembly Cloning Mix (Monad, Suzhou, China) and transformed into *Escherichia coli* DH5α via heat excitation. *Saccharomyces cerevisiae* strains INVSC1 were further transformed with the pYES2-NTB-*MYB8*, pYES2-NTB-*MYB15*, and pYES2-NTB-*MYB64* recombinant vectors using the Yeast Transformation Kit (Coolaber, Beijing, China) [31]. To assess the growth of these transformants, the experimental group (positive clones containing pYES2-NTB-*MYB8*, pYES2-NTB-*MYB15*, and pYES2-NTB-*MYB64*), as well as the negative control (pYES2-NTB vectors) were incubated in SG-U liquid medium containing different concentrations of PEG 3350 (0, 30, 60, 90, 120, and 135 mM). The cultures were incubated for 3–4 days with shaking, after which the cultures were diluted at 1, 10^−1^, and 10^−2^ ratios, spotted onto SG-U solid medium, and incubated again at 30 °C for 3–4 days. Finally, the grown colonies were observed and photographed for documentation.

### 4.8. Subcellular Localization Analysis

To investigate the subcellular localization of *SsMYB8*, *SsMYB15*, and *SsMYB64*, the CDS sequences of the three genes with the terminator codon removed were amplified using homology arm primers with restriction endonuclease cleavage sites *NcoI* and *SpeI*, respectively. The *SsMYB8*, *SsMYB15*, and *SsMYB64* genes were ligated into pCAMBIA1302 (containing a green fluorescent protein (GFP) tag) vector using MonCloneTM Single Assembly Cloning Mix (Monad, Suzhou, China) and transformed into *Escherichia coli* DH5α. The recombinant vector and empty vector were then transiently transformed into tobacco leaf cells. The transgenic tobacco plants were incubated in dim light for 3 d and observed by laser confocal microscopy. For co-localization, the marker plasmid (pCAMBIA1302- *ASH2L*, containing a red fluorescent protein (RFP) tag) was transformed into *Agrobacterium* EHA105, which was suspended with pCAMBIA1302-*SsMYB8*, pCAMBIA1302- *SsMYB15*, and pCAMBIA1302-*SsMYB64* plasmids, respectively, mixed 1:1 before injection, and then injected into tobacco leaves. The excitation wavelengths of green fluorescent protein (GFP) and red fluorescent protein (RFP) were 488 nm and 555 nm, respectively. Sequence primer information is provided in Appendix A.

## 5. Conclusions

In this study, we identified 113 members of the *MYB* gene family within the Sudan grass genome. These genes were classified into three groups based on their phylogenetic relationships with *Arabidopsis* and sorghum MYBs. Our structural analyses, encompassing gene structure, motif composition, and motif analysis, indicated that SsMYB proteins exhibit a notable level of conservation within specific subfamilies. The presence of diverse cis-regulatory elements in the promoter regions of *SsMYBs*, involved in developmental regulation, hormonal responses, and stress responses, likely underpins the multifaceted regulatory functions of these genes. Through collinearity analysis, we observed that the expansion of the *SsMYB* gene family predominantly occurred via segmental duplication. Furthermore, the duplicated gene pairs displayed distinct expression patterns, suggesting functional diversification within the gene family. Furthermore, interaction network analyses revealed possible interactions among several *SsMYB* genes, with *SsMYB15* and *SsMYB65* functioning as central hub genes. KEGG enrichment analyses showed that *SsMYBs* were mainly enriched in pathways such as amino acid synthesis. Yeast heterologous expression analysis showed that *SsMYB8*, *SsMYB15*, and *SsMYB64* played a positive regulatory role in drought stress response. Subcellular localization results reveal that *SsMYB8*, *SsMYB15*, and *SsMYB64* are all in the nucleus. These findings provide a basis for future studies on the regulatory functions of MYB proteins in pasture grass under abiotic stress.

## Figures and Tables

**Figure 1 plants-13-02645-f001:**
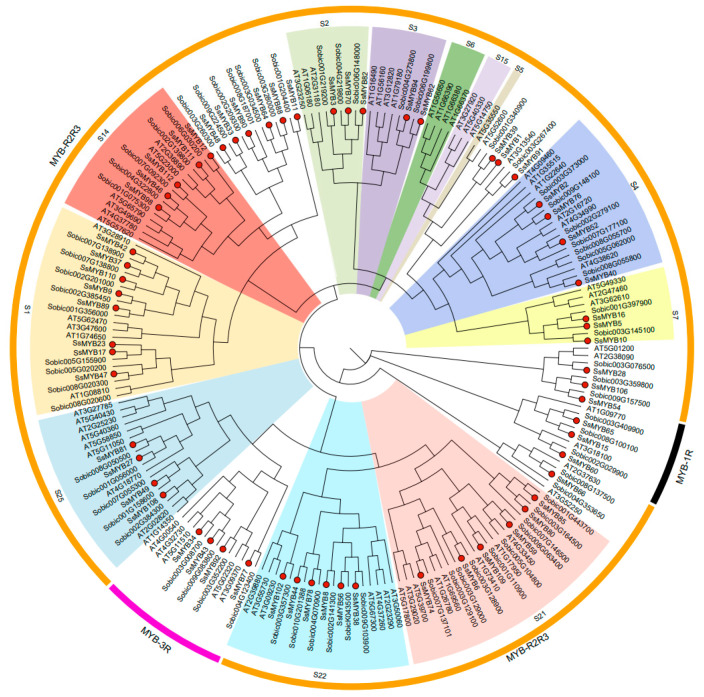
Phylogenetic analysis of MYB proteins in Sudan grass, *Arabidopsis thaliana*, and *Sorghum bicolor*. Different MYB subgroups (1R-MYB, R2R3-MYB, and 3R-MYB) are indicated in different colors. Red dots represent members of the Sudan grass MYB family, with each subgroup represented by a different colored block. The subclasses were designated as previously reported [18]. S1–S25 represent different subclasses of MYBs in *Arabidopsis* and classify *SsMYBs* of the same branch into the same subclass. The phylogenetic tree was constructed using the neighbor-joining (NJ) method with a bootstrap value of 1000.

**Figure 2 plants-13-02645-f002:**
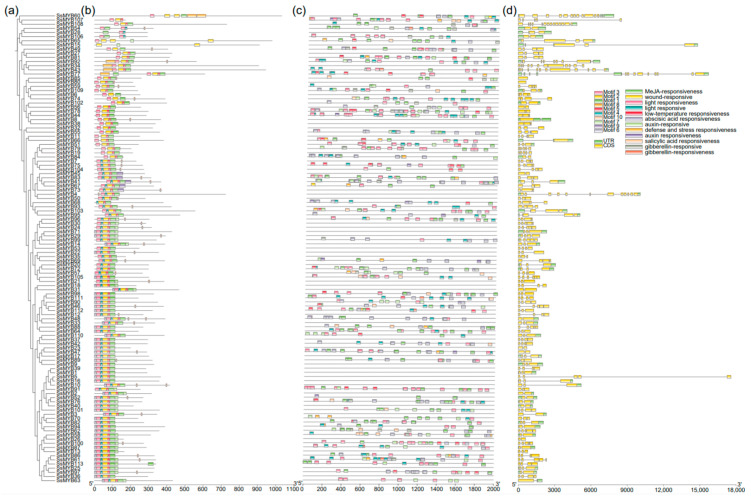
Phylogenetic tree, motif pattern, and gene structure analysis of the *MYB* gene family in Sudan grass. (**a**) Phylogenetic tree of the Sudan grass *MYB* gene family. (**b**) The 10 conserved MYB proteins are represented by different colored squares. (**c**) Characterization of the cis-acting elements in the promoter region of the *SsMYB* gene. (**d**) Exon-intron structure of the Sudan grass MYB proteins. The green frame represents the gene untranslated region (UTR), the yellow frame represents the gene exons, and the black lines represent the introns.

**Figure 3 plants-13-02645-f003:**
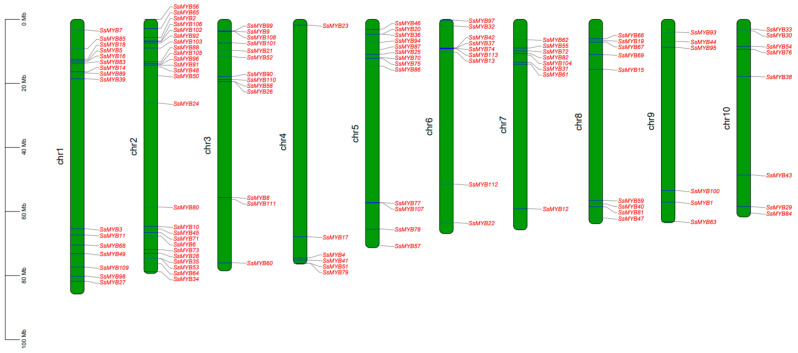
Distribution of 113 *SsMYB* genes on 10 chromosomes. The scale indicates chromosome length.

**Figure 4 plants-13-02645-f004:**
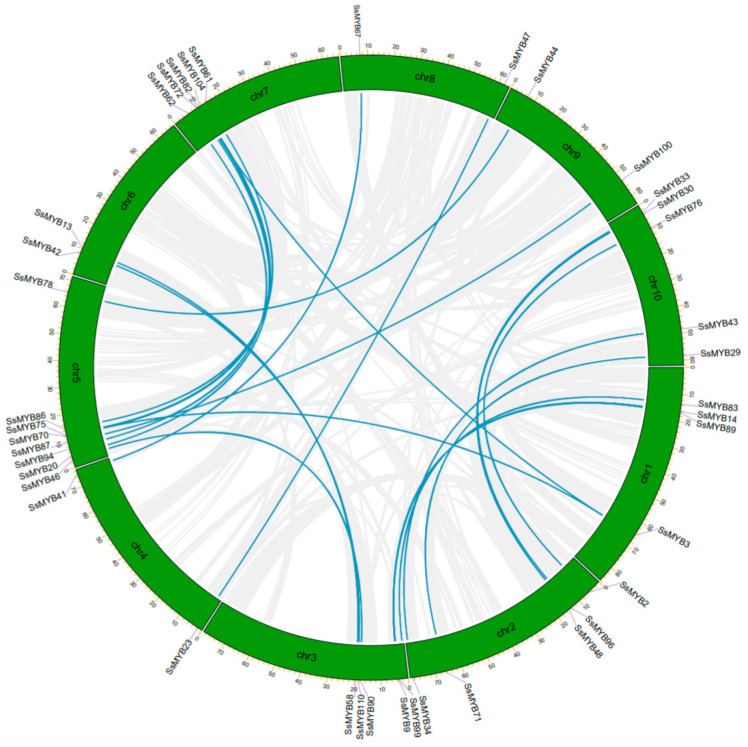
Distribution and collinearity analysis of MYBs in the Sudan grass genome. The blue line indicates the collinearity between the *SsMYBs*.

**Figure 5 plants-13-02645-f005:**
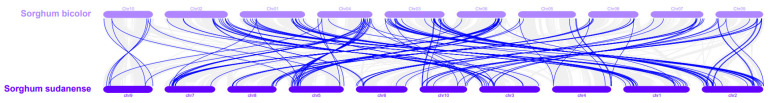
Collinearity analysis of *MYB* genes between Sudan grass and sorghum. Gray lines represent covariance blocks between the Sudan grass and sorghum, and blue lines represent covariant *MYB* gene pairs.

**Figure 6 plants-13-02645-f006:**
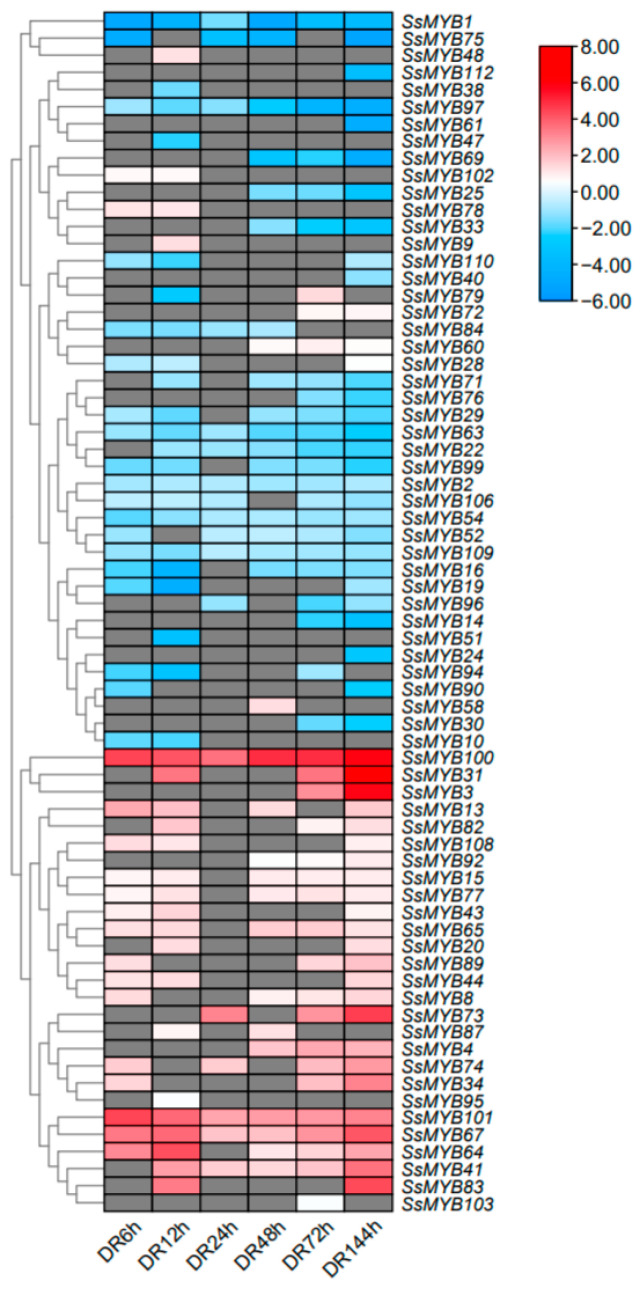
Expression levels of 70 *SsMYB* genes in the aboveground plant parts in response to drought stress. Gray frames indicate a lack of gene expression in the transcriptome data. DR: drought response; drought treatments: 6 h, 12 h, 24 h, 48 h, 72 h, 144 h.

**Figure 7 plants-13-02645-f007:**
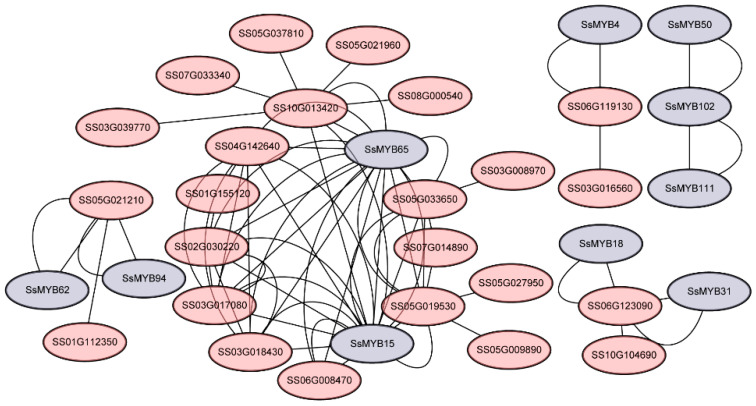
The interaction networks of 74 SsMYB proteins predicted based on the STRING database. The gray circles represent members of the *SsMYB* gene family, and the red circles represent other genes in Sudan grass.

**Figure 8 plants-13-02645-f008:**
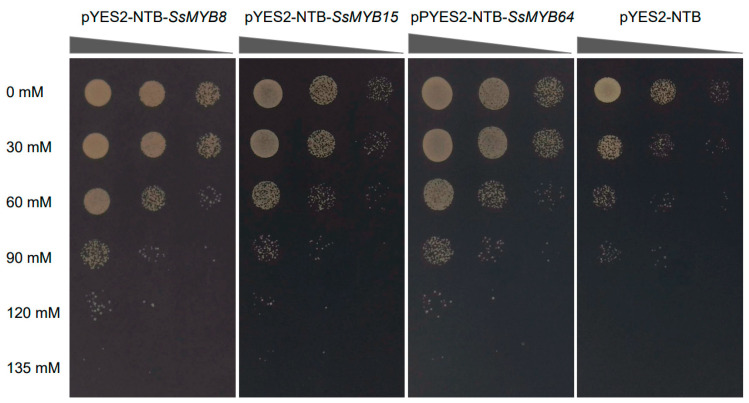
Overexpression of *SsMYB8*, *SsMYB15*, and *SsMYB64* improves drought tolerance in yeast. The growth of INSVC1 yeast transformed with pYES2-NTB containing *SsMYB8*, *SsMYB15*, and *SsMYB64* and empty vector pYES2-NTB. The left side indicates the concentration of different PEG3350 in SG-U medium. The top triangles indicate the OD values of the yeast, diluted tenfold as a gradient.

**Figure 9 plants-13-02645-f009:**
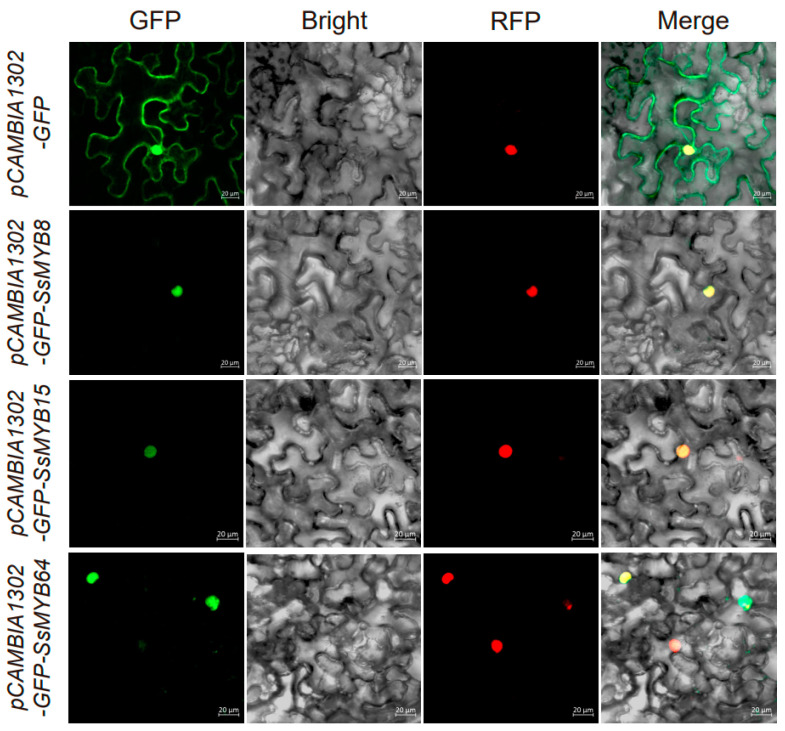
Subcellular localization of pCAMBIA1302-GFP-*SsMYB8*, pCAMBIA1302-GFP-*SsMYB15*, and pCAMBIA1302-GFP-*SsMYB64* fusion proteins in leaf epidermal cells of *Tobacco benthamiana*. Leaf epidermal cells transformed with pCAMBIA1302-GFP were used as controls. Scale bar: 20 μm. Primers used for vector construction are listed in Appendix A.

## Data Availability

The data presented in this study are available in the article and Appendix A.

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
