# Peer review of "Genome-Wide Identification and Characterization of MYB Transcription Factors in Sudan Grass under Drought Stress"

_plants, 2024, doi:10.3390/plants13182645_

Round 1

Reviewer 1 Report

Comments and Suggestions for Authors

In their manuscript, Liu et al. have analyzed the MYB transcription factor family in Sudan grass using various bioinformatic tools. The work is largely theoretical in nature and therefore of limited practical value. Little effort was made to experimentally confirm the computational findings. However, the authors have attempted to comprehensively describe MYB genes in a new grass species and provided an initial analysis. These results may be useful for future research in the field, which may justifies the manuscript's publication. Nevertheless, the manuscript requires substantial revision before it can be published. Below are some recommendations for improving the manuscript.

Major Comments:

  1. Title: The title is not appropriate. The authors did not functionally validate any MYB genes; they only demonstrated, using a heterologous system, that three of the MYBs could be involved in drought stress.
  2. Results:
    • Subchapter 2.1: Please specify the methods used for the identification and characterization of MYB genes, including their localization and other parameters.
    • Subchapter 2.2: I completely disagree with the interpretation of the phylogenetic tree (the same applies to this section in the discussion). The MYB-R2R3 genes definitely do not belong to a single cluster as stated but are instead distributed across several clusters. Please review the literature on phylogenetic trees and revise this subchapter accordingly.
    • Subchapter 2.3: “We categorized the MYB-DNA binding domain of SsMYB into four subdomains (Fig. S1).” Figure S1 is too small to evaluate this statement. Please provide a higher-resolution figure with improved readability and quality. Additionally, it is unclear how these subdomains were classified.
    • Subchapter 2.4: The information about motifs (“motifs 1, 2, 3” and so on) is unclear. How were these motifs selected? How are they conserved relative to each other? Are they present in all MYB genes, or only in a subset? This subchapter requires a clearer and more detailed description.
    • Subchapter 2.6: Are the transcriptomic data derived from another publication? If so, a reference is missing. Additionally, I do not see “decreasing/inversing trends” in the expression. The expression levels were simply increased or decreased. Please revise this wording. What threshold was used to distinguish differentially expressed genes from those with unchanged expression?
    • Subchapter 2.7: The description of the control and experimental groups is unclear. This section requires clarification and improvement. Furthermore, the criteria for selecting the three genes for analysis are not specified.
  3. Discussion: The discussion is weak and does not place the new information in the context of existing data. Some parts of the discussion (e.g., lines 299-311) repeat the results almost verbatim. Please rewrite the discussion to provide a more focused analysis.
  4. Figures:
    • Figure 1: The description of the figure is inadequate. What do the different colors in the dendrogram represent? What do the red points in the dendrogram indicate?
    • Figure 2: The subfigures are too small and barely visible. Consider a better presentation format. Additionally, the descriptions of subfigures (c) and (d) in the legend below appear to be swapped. The description of these subfigures is also incomplete. Legends explaining different motifs should be placed close to the corresponding subfigure for better understanding.
    • Figure 6: What do the abbreviations below the figure mean?
    • Figure 7: What criteria were used to label the genes in grey and red? In my view, some genes, such as MYB50, 102, and 111, are not hub genes.
    • Figure 8: The figure description is not appropriate. What do the triangles on the top represent? What do the concentrations on the left indicate?
  5. Table S1: What does "PPT" stand for?
Comments on the Quality of English Language

my comments regarding English in the main review

Reviewer 2 Report

Comments and Suggestions for Authors

It is a well structured paper about genome analysis and of MYB transcription factors in Sudan grass. This study is well-written, contains solid experimental analysis, conclusions are supported by results. The manuscript may benefit from following optimizations:

Line 138-139 - pleas specify if the statement that the motifs are conserved was derived from the literature (reference?) or from the actual bioinformatics analysis in this study

Line 161 - SsMYB is written sometimes in italics, sometimes not - please check throughout the text

Line 213-214 - were biosynthetic pathways for amino acid derivatives like polyamines (e. g. putrescine) also impacted?

Line 304-306 - may enrichment of the amino acid biosynthesis pathways had an indirect positive impact on secondary metabolism?

Comments on the Quality of English Language

Minor proof reading recommended 

Round 2

Reviewer 1 Report

Comments and Suggestions for Authors

In the resubmitted version, Liu et al. have presented an improved manuscript that addresses most of the reviewers' concerns. Notably, they have added a new chapter to the Results section, which was not present in the previous version. Despite these improvements, the manuscript is still not ready for publication for the following reasons:

The most significant issue is the description of the method for analyzing subcellular localization, as presented in Methods and Materials section 4.8. In general, methods should be detailed enough to allow replication by other researchers, which is not the case in this manuscript. For example:

What protein was used for the visualization of the fused product (I assume this was GFP)? How were the cDNAs cloned? Were they cloned in-frame? Were the fragments amplified for cloning? If so, what primers were used? Were the fragment ends modified for cloning? What protein (construct) was used for the visualization of the nucleus? What is meant by "marker plasmid"?

Additionally, the figure legends are still incomplete. Figures should be understandable based on their content and corresponding legends alone, which is not the case for many figures in the manuscript. A legend should start with a (sub-)figure title and explain all features of the figure. Specifically:

Figure 1: What do the abbreviations S1-S25 represent? Many genes do not seem to belong to any subgroups. Do these genes form their own subgroups?

Figure 2c: The description should begin with a subtitle such as “Exon-intron structure of the Sudan grass MYB proteins” followed by the existing text.

Legend to Figure 8: The sentence starting with “Overexpression of SsMYB8…” should be used as the title and moved to the beginning of the legend.

Smaller issue:

Line 159: “…the 113 MYB genes had 0-14 introns and 1-14 exons” should be revised to “…the 113 MYB genes had 0-13 introns and 1-14 exons.”

Comments on the Quality of English Language

no comments
